# Cluster-Segregate-Perturb (CSP): A Model-agnostic Explainability Pipeline for Spatiotemporal Land Surface Forecasting Models

## Abstract

Satellite images are increasingly valuable for modeling regional climate change. Earth surface forecasting is one task that combines satellite imagery and meteorological data to understand how climate evolves over time. However, understanding the complex relationship between meteorological variables and land surface changes remains a challenge. Our paper introduces a pipeline that integrates principles from perturbation-based techniques like LIME and global explainability techniques methods like PDP, addressing the limitations of these techniques in high-dimensional spatiotemporal models. This pipeline facilitates analyses such as marginal sensitivity, correlation, and lag analysis, etc for complex land forecasting models. Using ConvLSTM for surface forecasting, we analyzed influence of variables like temperature, pressure, and precipitation on the NDVI of the surface predictions. Our study in EarthNet2021 Dataset (primarily consists of samples from the European Alps region, collected during the spring to fall seasons) revealed that precipitation had the greatest impact, followed by temperature, while pressure has little to no direct effect on NDVI. Additionally, interesting nonlinear correlations between meteorological variables and NDVI have been uncovered.

## 1 Introduction

The Earth's climate is undergoing a significant transformation, posing a substantial threat to human existence. Its detrimental effects on terrestrial surfaces, which sustain most life on our planet, are becoming increasingly evident. From the depletion of Arctic sea ice Shokr & Ye (2023) to the intensification of fire incidents Lampe et al. (2023), the repercussions of climate change manifest across diverse and variable geographic regions. Studying the effects of meteorological variables like temperature, precipitation, and pressure is central to climate change analysis.

Over the past decade, there has been a notable increase in satellite sensors, leading to the availability of Earth observation data on an unprecedented scale. Initiatives like the Copernicus program Geudtner et al. (2014) offer high-resolution data with enhanced temporal coverage, enabling the generation of dense predictions and analyses that were previously unattainable. Land surface forecasting using spatiotemporal forecasting models plays a crucial role in predicting changes in surface conditions over time, such as vegetation growth Beyer et al. (2023) van Oorschot et al. (2023), soil moisture levels Chrysanthopoulos et al. (2023) Zhang et al. (2023), and land use patterns Witjes et al. (2022) Wang et al. (2023). However, interpreting the outputs of these models and understanding the factors driving their predictions pose significant challenges, particularly in the context of complex spatiotemporal data and high-dimensional feature spaces.

Explainability has emerged as a critical requirement for ensuring the transparency, trustworthiness, and reliability of machine learning models, including those used in land surface forecasting. Traditional explainability techniques, such as Local Interpretable Model-agnostic Explanations (LIME) Ribeiro et al. (2016) and partial dependence plots (PDP) Friedman (2001) etc, have limitations when applied to land surface forecasting models. These models often operate in high-dimensional spatiotemporal feature spaces, making it challenging to isolate the effects of individual variables on model predictions.

We introduce the Cluster-Segregate-Perturb (CSP) pipeline, a novel approach to explainability in land surface forecasting models. CSP pipelines clustering, segregation, and perturbation to enable comprehensive investigative analyses of model predictions. By clustering instances of meteorological variables, we identify unique patterns within each meteorological variable. After clustering, the data samples are segregated based on the earlier clusters. This segregation organizes the data into segments we termed, weather segments, representing smaller, more homogeneous subsets of the dataset based on distinct sets of meteorological conditions. Finally, by perturbing the meteorological variables of data samples within each segment, the pipeline creates artificial samples representing variations from the original data. These artificial samples make the analyses more robust.

In this paper, we present the development and evaluation of the CSP pipeline for investigative analyses on land surface forecasting models. We demonstrate its effectiveness through empirical evaluations in uncovering the relationships between meteorological variables and land surface evolution via NDVI.

## 2 Challenges in Explainability of Spatiotemporal Land Surface Forecasting Models

- The primary challenge lies in managing the temporal dynamics of meteorological variables. Unlike global explainability techniques like PDP, which use simple artificial samples to assess the marginal effects of features, generating realistic high-dimensional spatiotemporal data is complex due to the curse of dimensionality. Even if natural-looking spatiotemporal samples are created, the reliability of a pre-trained model's predictions can be compromised if it has not been trained on data reflecting similar patterns. This issue is prevalent in complex models with high-dimensional latent spaces, where approximating the distribution is often sparse and difficult.
  To address these challenges, we applied perturbations to the original training data, keeping them within natural weather patterns. This approach generated realistic data points, allowing the model to make robust predictions without the need for creating artificial samples from scratch.

- Another challenge is handling weather-based variations. As we know, within a season itself, different temperature, pressure, and precipitation patterns will affect the land surface evolution differently. Therefore, any analysis without properly segregating these natural weather patterns might yield incorrect aggregated values.
  To address this limitation, we divided the data into weather segments—smaller, more homogeneous subsets based on specific meteorological conditions. This approach allowed us to better understand the model's behavior across different weather scenarios, similar to LIME, but operating at the segment level instead of the sample level.

## 3 Related Work

### 3.1 Explainable AI

Explainable AI (XAI) techniques have been developed to address the lack of interpretability in deep learning models, including Convolutional Neural Networks (CNNs) and LSTM models. These techniques aim to provide insights into the decision-making process of these models. One approach is to use saliency maps, feature attribution, and local interpretable model-agnostic explanations (LIME) to enhance transparency and trustworthiness, another approach involves modifying the architecture of CNNs to improve interoperability, such as in Habib et al. (2022) Habib et al. uses sinc-convolution layers and explanation vectors to identify domain-specific insights and in De la Fuente et al. (2023) authors proposed a modification of the LSTM architecture called HydroLSTM, which enhances the interpretability of the model by representing internal system processes in a manner analogous to a hydrological reservoir.

Quantifying the global relationship between input features and model predictions in time-series image forecasting presents a significant challenge and remains an active area of research and the attention given to model explainability in time-series applications has not been as significant as in the fields of computer vision or natural language processing Rojat et al. (2021). Huang et al. in

Huang et al. (2022) suggest two techniques for explainability in the spatiotemporal predictive learning task (SPLT): first, the synthesis of multiple independent components to analyze how the features contribute to the prediction; and second, a state decomposition and expansion technique to separate intertwined signals in the spatiotemporal dynamical system helping in exploring the mechanisms underlying motion formation. they concluded that a collaboration mechanism, namely, extending the present and erasing the past (EPEP), explains the motion formation in SPLT. Duckham et al. in Duckham et al. (2022) utilises the Simple Event Model and PROV-O ontologies to enable queries not just about reasoner inferences but also about explanations for specific conclusions reached by the system. This capability is embedded in the NEXUS system, which combines multiple reasoning components that can support a wide range of spatiotemporal queries. Moosburger in Wang (2023) proposed an interpretable and modular framework for unsupervised and weakly-supervised probabilistic topic modelling of time-varying data. This framework merges generative statistical models with computational geometric techniques. Pham et al. in Pham et al. (2023) proposed Temporally Weighted Spatiotemporal Explainable Neural Network for Multivariate Time Series (TSEM) merges RNN and CNN capabilities by using RNN hidden units to weigh the temporal axis of CNN feature maps. It matches STAM's accuracy and fulfils several interpretability standards, including causality, fidelity, and spatiotemporality.

### 3.2 EXPLORING METEOROLOGICAL VARIABLES THROUGH NDVI ANALYSIS

Several studies have examined the sensitivity, correlation, lag, etc of meteorological variables such as precipitation and temperature concerning NDVI. Li and Guo, in Li & Guo (2010), analyzed the response characteristics and sensitivity of NDVI to climatic factors in Tianjin, China. They found that NDVI increases with rising temperatures but gradually decreases with increasing precipitation. Moreover, the impact of temperature on NDVI was more pronounced in spring and autumn, while precipitation had a dominant effect in spring, autumn, and early summer. Shun et al. Pan et al. (2019), concluded that NDVI exhibits a higher correlation with air temperature in high-altitude alpine and plateau areas, whereas it correlates more strongly with precipitation in grassland and desert grassland regions. Yujie et al. in Yang et al. (2019), concluded that precipitation was identified as the most significant factor affecting vegetation evolution, followed by temperature, land cover change, population, elevation, and nightlight. Hao et al. Hao et al. (2012) studied the link between climatic variables and NDVI in the upper stream of the Yellow River and a strong 'correlation between NDVI and precipitation for grassland and forest. Their results suggest that higher precipitation levels lead to elevated NDVI values. Additionally, the monthly highest temperature and precipitation significantly affected NDVI. Similarly, Wang et al. in Wang et al. (2001) explored the spatial distribution and year-to-year changes in NDVI across the central Great Plains and found a clear link between NDVI patterns and average annual precipitation. A strong correlation was observed between NDVI and precipitation deviations during dry years, like 1989, following another dry year. Conversely, a weak correlation was observed during wet years, such as 1993, one of the wettest on record. These findings indicate higher correlation coefficients during or after dry periods. In Feng et al. (2021), Jianming et al. . studied the time accumulation effect of meteorological variables on NDVI. They observed a positive correlation between NDVI and accumulated temperature, accumulated precipitation, and effective accumulated precipitation.

## 4 ASSUMPTION

- **Input features are independent of each other.** In marginal feature analysis, feature independence simplifies the process by isolating the effect of one variable on the outcome while holding all others constant. This ensures that changes in one feature do not influence the distribution or behaviour of other features, leading to a clearer understanding of their relationship with the outcome.

## 5 CSP PIPELINE

- **Cluster** Identify the inherent and distinct spatiotemporal patterns within each input feature by clustering.
- **Segregate** Partition the samples into segments containing smaller, more homogeneous subsets of the dataset based on distinct sets of spatiotemporal patterns of the features, By doing

this, any analysis performed on these groups reflects the behaviour of a specific local environment, allowing for more focused and insightful interpretations.

- **Perturb** Perturbation creates artificial samples in a segment. This facilitates many exploratory studies besides sensitivity analysis and makes the aggregated values more robust.

To demonstrate the application of the CSP pipeline to a land surface forecasting model, we employed a ConvLSTM model trained on the EarthNet21 dataset. It's worth noting that the pipeline is model-agnostic, meaning it can be seamlessly integrated with any other model architecture.

## 5.1 DATASET, MODEL AND METRIC OVERVIEW

### 5.1.1 EARTHNET2021 - DATASET

EarthNet2021 Requena-Mesa et al. (2021) is a large-scale dataset and challenge for Earth surface forecasting, which involves predicting satellite imagery conditioned on future weather.

The Dataset consists of 32,000 samples within the European region, each comprising a series of 30 Sentinel-2 images, each captured at intervals of 5 days. These images contain four bands (red, green, blue, and near-infrared) with a spatial resolution of 128x128px or 2.56 km$^2$ and a ground resolution of 20m. Moreover, accompanying weather-related meteorological data is included, such as precipitation, sea level pressure, and temperature (minimum, maximum, and mean), each comprising a series of 150 images for 150 days, at a coarser spatial resolution of 80x80px or 102.4 km$^2$, sourced from the observational dataset E-OBS Haylock et al. (2008).

Before the model training, as a preprocessing step, the spatiotemporal resolution of meteorological variables is matched to that of the Sentinel-2 images.

This study used the IID (In-Domain) train set fraction of the EarthNet2021 dataset denoted by $D$ containing 23904 samples denoted by $N$ in our analyses.

$$D = \{x_1, \ x_2, \ x_3, \ \dots \ x_N\}$$
$$x_i = (T_{\text{avg}_i}, \ T_{\min_i}, \ T_{\max_i}, \ P_i, \ R_i) \tag{1}$$

$x_i$ is a data sample $T_{\text{avg}_i}, \ T_{\min_i}, \ T_{\max_i}, \ P_i, \ R_i$ are the 30 timesteps spatiotemporal channels representing average temperature, minimum temperature, maximum temperature, pressure and precipitation respectively each $\in \mathbb{R}^{(30*128*128)}$. Additionally, other channels representing $DEM, red, blue, green, near-infrared, cloud mask, scene classification label$, and $data quality mask$ are also utilized during model training and inference. However, since they are not the main focus of this study, they are not explicitly notated.

### 5.1.2 CONVLSTM

ConvLSTM (Convolutional Long Short-Term Memory) networks have been proposed as effective methods for remote sensing time series analysis. These networks leverage the temporal and spatial contextual information present in time series images to improve classification accuracy. It was first used for precipitation nowcasting in Shi et al. (2015) since then it has been used for tasks such as land cover classification, change detection, and time series reconstruction.

In this study, we utilized the ConvLSTM model, as described by Diaconu et al. Diaconu et al. (2022). We trained the model on the IID (In-Domain) split of the EarthNet2021 dataset for 60 epochs. During this training phase, we attained an EarthNetScore of 0.3257, closely aligning with the score reported in the original paper, 0.3266. Here the EarthNetScore (ENS) is a composite evaluation metric used for assessing the performance of Earth surface prediction models it is the harmonic mean of the four components (MAD, OLS, EMD, SSIM), scaled between 0 (worst) and 1 (best) as described in Requena-Mesa et al. (2021).

Let $M_\theta$ denote the ConvLSTM model having $\theta$ as pre-trained weights. then the prediction $y_1$ is equated as:

$$y_i = M_\theta(x_i) \tag{2}$$

here $y_i = (r_i, \ g_i, \ b_i, \ nir_i)$, where $r_i, g_i, b_i$ and $nir_i$ represent the red, blue, green and near-infrared channels of the output respectively, and each channel $\in \mathbb{R}^{(20*128*128)}$.

### 5.1.3 SOFT-DTW (SOFT DYNAMIC TIME WARPING)

Soft-DTW Cuturi & Blondel (2017) is an extension of Dynamic Time Warping, which is a method for measuring similarity between two sequences that may vary in time or speed. DTW is particularly useful when comparing time series data where there might be temporal distortions or differences in the pacing of events.

In our investigation, soft-DTW served as the distance metric within k-means for clustering meteorological time series variables. For this purpose, we employed tslearn, a Python library specialized in time series data analysis, featuring pre-implemented soft-DTW functionalities Tavenard et al. (2020). The formal definition of soft-DTW :

$$soft-DTW^{\gamma}(\tau, \tau') = \min{}_{\pi \in \phi(\tau, \tau')}^{\gamma} \sum_{(i,j) \in \pi} f(\tau_i, \tau_j')^2 \qquad (3)$$

$$\min{}^{\gamma}(q_1, \ldots, q_n) = -\gamma \log \sum_i e^{-q_i/\gamma} \qquad (4)$$

Here $\min^{\gamma}$ is the soft-min operator parametrized by a smoothing factor $\gamma$ this makes it differentiable everywhere. $\phi(\tau, \tau')$ represents the set of all possible alignments between the two input sequences $\tau$ and $\tau'$. This set contains all the possible ways the elements of the two sequences can be aligned to each other. Each alignment, denoted by $\pi$, is a set of pairs $(i, j)$ where $i$ is an index from sequence $\tau$ and $j$ is an index from sequence $\tau'$. These pairs represent the correspondences between elements of the two sequences in a particular alignment. $f(\tau_i, \tau_j')$ represents the distance or dissimilarity between the $i$-th element of sequence $\tau$ and the $j$-th element of sequence $\tau'$, in eq 4 it is denoted as $q_i$. In the case of tslearn Tavenard et al. (2020), this distance of dissimilarity is Euclidean distance.

### 5.2 APPLYING THE STAGES OF THE CSP PIPELINE

#### 5.2.1 CLUSTERING THE METEOROLOGICAL VARIABLES

In the EarthNet21 dataset, each sample covers a small geographical area of only 2.56 km$^2$, resulting in minimal variability of meteorological variables like temperature, precipitation, and pressure across the spatial resolution at any given time step. Leveraging this characteristic, we simplified the clustering process from spatiotemporal to temporal clustering. This involved downsampling the meteorological variables to a single value for each time step, followed by time-series k-means clustering using soft-DTW on the downsampled meteorological variables.

To downsample the meteorological variables of each data sample we computed the average of the pixel values of each timestep image. The spatial resolution of each image is (128,128) px representing 2.56 km$^2$ ground resolution, we average it down to a single value. This approach condenses the meteorological variables to 30 values each representing one timestep. This downsampling is justified as the pixel values of these variables exhibit minimal variations within the channel images, and this even aligns with the amount of variance of a meteorological variable in a small geographical area.

Following the downsampling we conducted k-means clustering on each meteorological variable across different cluster sizes and obtained the optimal cluster size $K$ by selecting the one with the highest cluster $GoodnessScore$.

$$GoodnessScore_K = \frac{interCentroidScore_K}{intraClusterScore_K} \qquad (5)$$

We performed clustering for cluster sizes $\in \{2, 3 \ldots 15\}$. The goal is to identify a set of unique cluster centroids for each meteorological variable exhibiting a low $intraClusterScore$ and a high $interCentroidScore$.

$$intraClusterScore_K = \sum_{i=1}^{K} \frac{\sum_{j \in d_i} e^{DTW(centroid_i^K, j)} N_i}{N} \qquad (6)$$

After clustering for a specific cluster size $K$, $d_i$ is the subset of samples assigned to the cluster with centroid $centroid_i^K$ where $i \in \{1, 2 \ldots K\}$. In Equation 6, we calculate the fractional average exponential DTW value between the centroid and the samples assigned to that centroid where $\frac{N_i}{N}$ is

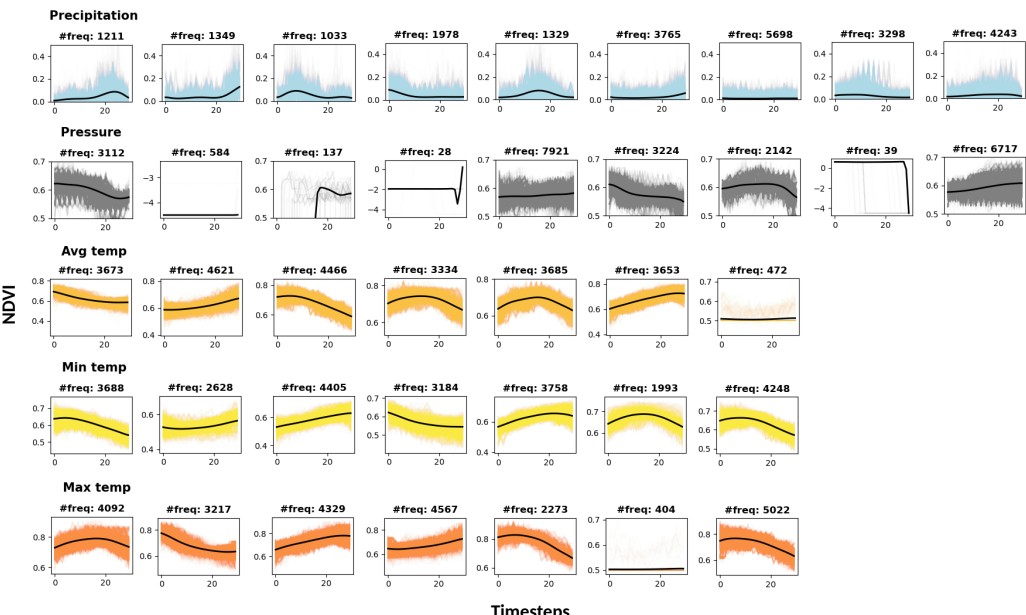

Figure 1: We've identified 9 unique clusters (representing base temporal patterns) for $precipitation$ and $pressure$, and 7 unique clusters for $temperature$. The $frequency$ displayed on each plot indicates the number of data samples assigned to each cluster. In the pressure column the $2^{nd}$, $4^{th}$ and $8^{th}$ plots exhibited noise patterns in the pressure data. Therefore, we can safely disregard these clusters.

the weight for the fractional average. $N$ refers to the total number of data samples, and $N_i$ refers to the count of samples belonging to cluster $i$. The function $DTW(x, y)$ is unbounded from above and its lower bound is zero. When the two temporal signals are more similar, the value of the function approaches zero so ultimately a lower value of $intraClusterScore$ is desirable.

$$interCentroidScore_K = \frac{CentroidPairs(K) - SimilarCentroids(\lambda)^2}{CentroidPairs(K)}$$

$$CentroidPairs(K) = \frac{K(K-1)}{2} \quad (7)$$

$$SimilarCentroids(\lambda) = \sum_{i=1}^{K} \sum_{i=j+1}^{K} \mathbb{1}\left(DTW(centroid_i, centroid_j) < \lambda\right)$$

For $interCentroidScore$ in the set of equations in 7 we penalise the cluster for having more similar centroids because we aim to find unique temporal patterns. Hence, a higher value of $interCentroidScore$ is desirable. Again we used the $DTW(x, y)$ function for calculating the similarity score, as discussed earlier this function is only bounded from below so we put a threshold $\lambda$ for classifying the centroids into two classes i.e., similar or sufficiently different to be deemed unique. through trial and error, we set its value to $0.4$.

Figure 1 illustrates the unique clusters identified for each meteorological variable.

After the clustering step, the prediction of the cluster-index $I_\alpha$ of a time series signal $Q$ is defined as

$$I_\alpha = m_\alpha^K(Q) \quad (8)$$

where $m_\alpha^K$ denote the k-means model with the highest $GoodnessScore$ having $K$ unique clusters for meteorological variable $\alpha$ where $\alpha$ is a $\in \{t_{avg}, t_{min}, t_{max}, p, r\}$ shortened notation for meteorological variables.

### 5.2.2 SEGREGATING SAMPLES INTO WEATHER SEGMENTS

We partitioned the dataset into segments, containing smaller, more homogeneous subsets of distinct temporal meteorological patterns. These segments are recognized by a tuple $c_i$ having five cluster-

index $I_\alpha$, one for each meteorological variable $\alpha$. Since a segment is a set of samples with similar weather conditions regardless of their geographical location, we refer to it as **weather segment** and notate this set as $S_i$.

$$c_i = (I_r, I_p, I_{t_{avg}}, I_{t_{min}}, I_{t_{max}}) \tag{9}$$

We identify the weather segment $S_i$ of a sample by determining the tuple $c_i$, by applying the k-means models in 8 to each meteorological variable of the sample.

$$S_i = \{x_j : (m_r^9(R_j), m_p^9(P_j), m_{t_{avg}}^7(T_{avg_j}), m_{t_{min}}^7(T_{min_j}), m_{t_{max}}^7(T_{max_j})) = c_i\} \tag{10}$$

Thus the entire dataset is segregated into distinct segments, $D = S_1 \cup S_2 \cup S_3 \ldots \cup S_{N_c}$ and through this process we uncovered a total of $N_c = 784$ weather segments.

### 5.2.3 PERTURBATION

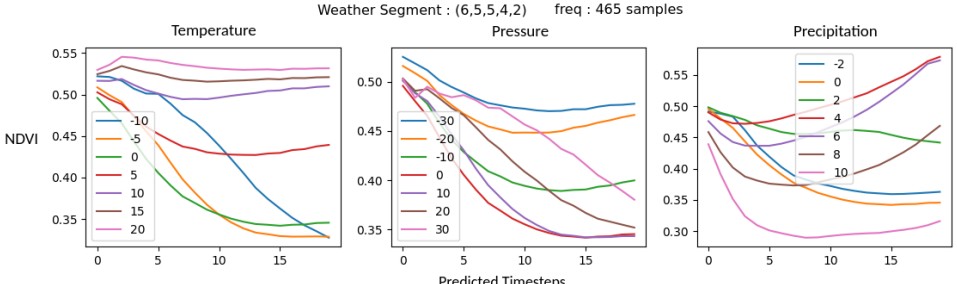

Figure 2: Illustrate average NDVI signals for different perturbations of the meteorological variables for a weather segment whose $c_i$ is (6,5,5,4,2), and frequency is 465 samples.

We created artificial samples by making small adjustments to one of the meteorological variables at a time while treating the others as constant across all the samples within a segment.

Since the samples originate from Central and Western Europe, primarily within the temperate zone, we have taken precautions to ensure that the variables remain within natural bounds. The average temperature of the dataset hovers around $20\,^\circ$C. Therefore, the additive adjustment for temperature $\delta_1 \in [-10, -5, 0, +5, +10, +15, +20]\,^\circ$C. Similarly, the average precipitation revolves around 1.5 mm/30 timesteps and the average pressure revolves around 1020 hPa so the additive adjustment in precipitation and pressure are $\delta_3 \in [-2, 0, +2, +4, +6, +8, +10]$ mm/30 timesteps and $\delta_2 \in [-30, -20, -10, 0, +10, +20, +30]$ hPa respectively. The curves in fig 2 show a weather segment's average perturbed NDVI signals.

Let $V^{(\delta_1, \delta_2, \delta_3)}$ be the small additive adjustment done to the meteorological variables, defined as:

$$V^{(\delta_1, \delta_2, \delta_3)} = (v_{t_{avg}}^{\delta_1}, v_{t_{min}}^{\delta_1}, v_{t_{max}}^{\delta_1}, v_p^{\delta_2}, v_r^{\delta_3})$$
$$v_\alpha^t = A_{ijs} \mid \forall i \forall j \forall s \{A_{ijs} = t\} \text{ and } \in \mathbb{R}^{(30*128*128)} \tag{11}$$

eq: 11 simply suggests that adjustment $V^{(\delta_1, \delta_2, \delta_3)}$ is a tuple similar in shape and order to a data sample defined in eq: 1. $\delta_1, \delta_2, \delta_3$ is the amount of additive adjustment for temperature, pressure and precipitation channels respectively.

Also, the meteorological channels in the EarthNet21 dataset have been normalized using the eq: 12 so we also normalized these adjustments $\delta_1, \delta_2, \delta_3$ before adding them to their respective meteorological channels.

$$R_{mm} = 50R, \quad P_{hPa} = 200P + 900, \quad T_{\circ C} = 50(2T - 1) \tag{12}$$

## 6 EXPERIMENTS

We conducted two investigative analyses: marginal sensitivity analysis and marginal correlation analysis between the meteorological variables and NDVI of the output of the ConvLSTM prediction.

We choose NDVI (Normalized Difference Vegetation Index) for this study since changes in meteorological variables like temperature, precipitation, and pressure greatly affect vegetation health and density.

$$NDVI\,(y_i) = \frac{nir_i - r_i}{nir_i + r_i} \tag{13}$$

### 6.1 MARGINAL SENSITIVITY ANALYSIS

We conducted the marginal sensitivity analysis of each meteorological variable on the NDVI of the predicted output within each weather segment. This analysis is localized because each weather segment contains only a subset of the samples thus following a distinct pattern of these meteorological variables.

The average NDVI signal of a segment labelled with tuple $c_i$ having the sample set $S_i$ corresponding to the perturbations $(\delta_1, \delta_2, \delta_3)$ in a meteorological variable is defined as:

$$\overline{NDVI}^{c_i}_{(\delta_1,\delta_2,\delta_3)} = \frac{\sum_{x \in S_i} NDVI\left(M_\theta(x + V^{(\delta_1,\delta_2,\delta_3)})\right)}{|S_i|} \tag{14}$$

The curves from the eq: 14 are visualized in fig:2.

Since we are doing marginal analysis, we set $\delta_2 = \delta_3 = 0$ when computing the above metric 14 for temperature. Similarly for pressure $\delta_1 = \delta_3 = 0$ and precipitation $\delta_1 = \delta_2 = 0$. Hence local marginal sensitivity of temperature($t_{avg}$), pressure($p$) and precipitation($r$) for the weather segment labeled with tuple $c_i$ is given as:

$$Senstivity^{c_i}_{t_{avg}} = \sum_a \sum_{b|a \neq b} \frac{\left|\overline{NDVI}^{c_i}_{(a,0,0)} - \overline{NDVI}^{c_i}_{(b,0,0)}\right|}{|a-b|}$$

$$Senstivity^{c_i}_{p} = \sum_a \sum_{b|a \neq b} \frac{\left|\overline{NDVI}^{c_i}_{(0,a,0)} - \overline{NDVI}^{c_i}_{(0,b,0)}\right|}{|a-b|} \tag{15}$$

$$Senstivity^{c_i}_{r} = \sum_a \sum_{b|a \neq b} \frac{\left|\overline{NDVI}^{c_i}_{(0,0,a)} - \overline{NDVI}^{c_i}_{(0,0,b)}\right|}{|a-b|}$$

After determining the marginal sensitivity of meteorological variables $\alpha \in \{t_{avg}, p, r\}$ for individual weather segments, we found out that irrespective of the weather segment the sensitivity of the variables remained almost the same so we approximated the global sensitivity by computing the weighted average where weights are the cardinality of the sample sets of each weather segment, denoted by $|S_i|$.

$$Senstivity_\alpha = \sum_{i=1}^{N_c} (Senstivity^{c_i}_\alpha * |S_i|) \tag{16}$$

RESULT

Table 1: Marginal NDVI sensitivity table

| Variable | Sensitivity | SD | Unit |
|---|---|---|---|
| Precipitation | 0.0183 | 0.0043 | per mm |
| Temperature | 0.0034 | 0.0014 | per °C |
| Pressure | 0.0015 | 0.0007 | per hPa |

Table 1 demonstrates that within the region of study i.e., Europe, a unit change in precipitation has the most significant effect on the NDVI value, followed by temperature and pressure in sequence. To quantify this impact, the weight of precipitation is approximately 12 times greater than that of pressure and roughly 5 times greater than that of temperature.

## 6.2 MARGINAL CORRELATION ANALYSIS

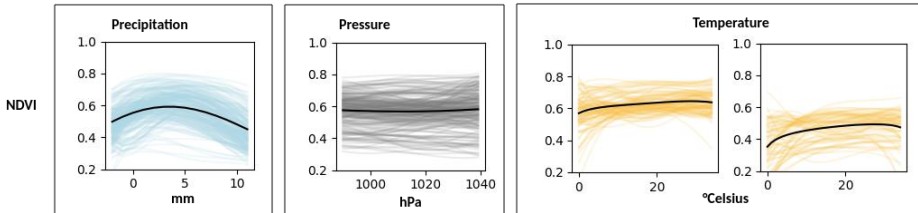

Figure 3: Correlation patterns between meteorological variables and NDVI of the predictions. *Temperature is split into two curves to enhance visualization: lower NDVI scenes exhibit greater correlation curve curvature, which decreases as NDVI increases.*

We conducted a correlation analysis between meteorological variables and NDVI of the predictions. We aimed to identify a best-fitting correlation curve for each meteorological variable $\alpha \in \{t_{avg}, p, r\}$ within each weather segment denoted by the tuple $c_i$. However, for curve fitting, we needed a set of points representing the aggregated curve for each meteorological variable $\alpha$ within a weather segment $c_i$:

$$Points_\alpha^{c_i} = \{(x_{b_1}, y_{b_1}), (x_{b_2}, y_{b_2}), \dots\} \tag{17}$$

Here $y_{b_i}$ is the median value obtained by downsampling the average NDVI signal of the weather segment $c_i$ for the perturbation $b_i$ i.e., downsampling the curve obtained through the eq: 14 and $x_{b_i}$ is the mean value of the meteorological variable $\alpha$ of the weather segment $c_i$ added to the perturbation $b_i$. Value of the perturbation $b_i \in \delta_1$ in case of temperature, $\in \delta_2$ in case of pressure and $\in \delta_3$ in case of precipitation as given in 11 and $\forall i \forall j \{b_i \neq b_j\}$.

With the points set in hand, we conducted curve fitting using a range of linear and non-linear models, including polynomial, exponential, logarithmic, sinusoidal, and Gaussian curves.

$$coeff_\alpha^{c_i} = fit(eq, Points_\alpha^{c_i}) \tag{18}$$

Here the $fit$ function takes a curve equation and a set of points as parameters and returns the coefficients of the best-fitting curve. It was discovered that second-degree polynomials gave the best fit for most weather segments across all meteorological variables.

We plotted the curves from all the weather segments to visualize the underlying pattern in 3. We standardized the range of $x_{a_i}$ for different meteorological variables i.e., for temperature, the range was standardized to $[0, 35]°C$ similarly for pressure and precipitation it was $[990, 1040]hPa$ and $[-2, 12]mm$ respectively. Finally, we used the best-fitted curve to calculate the approximated value of $y_{a_i}$ for the ranges of $x_{a_i}$ as mentioned above.

RESULT

Table 2: Correlation curves

| Curve | $a'$ | $b'$ | $c'$ |
|---|---|---|---|
| Precipitation | -0.0034 | 0.0252 | 0.5554 |
| Temp-$NDVI_{high}$ | 0.0001 | 0.0044 | 0.5805 |
| Temp-$NDVI_{low}$ | -0.0002 | 0.0096 | 0.3750 |
| Pressure | 2.18e-05 | -0.0441 | 22.8677 |

Table 2 illustrates the correlation curves. The analysis suggests that the correlations exhibit nonlinear behaviour and can be described with the equation of the parabolic curve:

$$y = a'x^2 + b'x + c'$$

. Figure 3 illustrates that increasing precipitation and temperature lead to higher NDVI values. However, with precipitation, this rise reaches a threshold somewhere around 4-5 mm, beyond which

NDVI declines. Notably, when NDVI is low, the correlation curves for both temperature and precipitation exhibit more pronounced curvature. This curvature diminishes as NDVI increases, suggesting a diminishing effect of additional precipitation on NDVI in scenes with already high vegetation indices. Additionally, the relationship between pressure and NDVI appears nearly linear, indicating the minimal impact of pressure changes on NDVI.

## 7 LIMITATIONS

While the CSP pipeline is simple, model-agnostic and inherently generic, the approach can have some limitations.

- The quality of results in the CSP pipeline depends heavily on selecting the appropriate clustering method. Additionally, the performance of these methods is highly sensitive to hyperparameter settings, such as the number of clusters or the choice of distance metric. Poorly tuned parameters can result in suboptimal clustering, which negatively impacts the overall analysis.

- Limited research in the field of Spatiotemporal clustering. In Ansari et al. (2020) Ansari et al. categorised spatiotemporal clustering into six broad categories however the review focuses more on individual spatiotemporal points, such as events, geo-referenced data items, time series, moving objects, and trajectories unlike which most land surface prediction dataset contains sequences of satellite images and meteorological data. For clustering entire spatiotemporal samples, the technique would need to be adjusted to emphasize the clustering of these samples as a whole, rather than focusing solely on the characteristics of individual points. Time complexity would be another crucial aspect to consider.

- Limited research in the field of perturbations for attaining meaningful transformations. Most existing techniques, such as adversarial perturbation, Gaussian noise injection, etc add random noise to enhance model's robustness. In contrast, the CSP pipeline requires perturbations for meaningful transformations tailored to specific analytical goals. For example, given a real state cost prediction model using satellite imagery, one might ask how the average cost of the scene changes when buildings become more compact or more green space is added. while techniques like Variational Autoencoders (VAEs) may also be beneficial, they can be difficult to train.

## 8 CONCLUSION AND FUTURE WORK

This paper presented a pipeline to improve the explainability of complex land surface prediction models like ConvLSTM. The proposed methodology enables various investigative analyses, enhancing our understanding of the relationship between meteorological variables and model predictions. Our analysis revealed that NDVI exhibits the highest marginal sensitivity against precipitation, followed by temperature and pressure, with approximate ratios of 12:2:1. Moreover, we observed a nonlinear correlation between NDVI and meteorological variables, resembling a parabolic curve. Furthermore, as the average NDVI of the scene increases, the influence of precipitation and temperature on the curvature of the correlation curve diminishes. Additionally, it is concluded that pressure has little to no direct effect on NDVI.

In the future, we aim to pursue further advanced studies. This involves exploring the transitional impacts of meteorological variables by using techniques like Perlin noise Perlin (1985) which can ensure smoother interpolations, alongside conducting lag analysis. Nonetheless, the challenges in the clustering process need to be researched more, It is imperative to cluster samples based on factors other than meteorological variables, such as crop type, elevation, building density, etc and downsampling might not be the option every time, to address this obstacle, we intend to develop a spatiotemporal deep clustering method, thus enhancing the methodology's adaptability to handle more diverse datasets.

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
