# OpenReview forum: "Cluster-Segregate-Perturb (CSP): A Model-agnostic Explainability Pipeline for Spatiotemporal Land Surface Forecasting Models"
_ICLR.cc/2025/Conference — ICLR 2025 Conference Withdrawn Submission_

### Official Review · Reviewer_tHE2 · 2024-10-27

**Soundness:** 1
**Presentation:** 2
**Contribution:** 2
**Rating:** 3
**Confidence:** 4

**Summary:**

The paper proposes a pipeline to quantify the importance of different input variables for a Convolutional LSTM. The authors construct a model that aims at predicting the NDVI in a chip, based on different meteorological input variables. Once this model is trained, the authors propose a pipeline called Clustering, Segregation, Perturbation (CSP), that aims at identifying the importance of the different meteorological variables used in the ConvLSTM.
The pipeline starts by clustering in time: using k-means, the method identifies clusters of observations that can be grouped together over time. The second step is to segregate the clusters by k-means. The last step is to perturb the inputs, and observe the impact of the perturbation on the model output.

**Strengths:**

The paper addresses a timely and important subject: can we understand how a certain variable affects what is observed in deep learning models, without explicitly modeling the physical phenomena. This topic is of utmost importance for deep learning, as they are often critiqued as black boxes. This paper thus proposes a method to disentangle this black box, and shed a light on the importance of each variable on the observed process.

**Weaknesses:**

It is unclear to me what the clustering process adds here. The proposed perturbation resembles a simple sensitivity analysis. In table 1, the results are presented as averages over the clusters, and in figure 3 the different clusters are shown individually, but it is hard to understand weather these results would be different if the perturbation had been performed without the clustering.

Towards the end, the paper aims at demonstrating the impact of the different variables, and states an approximatively 12 times greater importance of the precipitation compared to the pressure, and 5 times greater than temperature. This seems significatif, but hard to judge given that we are not presented with a baseline. Comparing this method to other well known methods of sensitivity analysis, like simple perturbation or noise injection could inform about the significance of the result.

Overall, the paper was hard to read. The paper is not very well structured, and the methodology hard to follow. Multiple specific mistakes (c.f. specific comments below) made it hard to make sense of the methodology.

# Specific comments

Line 28: what interesting nonlinear correlations

Line 74: what is the curse of dimensionality?

Line 151: assumption: is this reasonable? Environmental variables are known to be not completely independent of each other

Line 177: so a total of 30 * 32000 images?

line 179: The spatial resolution is 20 meters, the dimension of the image is 128x128 pixels. I'm guessing the value 2.56 is the side of the image, so km, not an actual surface (20 m * 128 px / 1000 m/km = 2.56 km, the surface would be 6.5536 km2).

Line 181: "150 images for 150 days", so 1 image per day?

Line 181: unclear what the spatial resolution is. Assuming 102.4 is the surface (unit is km2, but this might be the same error as above), then the spatial resolution is 126.5 meters. But if it's the side, as above, then the spatial resolution is 1.28 km, which is probably more reasonable for a weather variable. Although E-OBS has a 10km resolution, so that would still be different.

Line 183: What is the resampling procedure? Probably nearest neighbor for spatial, but what about temporal?
How many pixels cover the same area as the Sentinel-2 chip? Assuming a 126.5 meters resolution, 20 pixels cover the chip, but if we take the value 1.28 km resolution, then 2 pixels per side (4 total) cover the chip, which means all the pixels in the resampled product basically have the same value. Even less if we consider the E-OBS resolution of 10km.

Line 185-196: not clear

Line 206: I'm not familiar with the EarthNetScore, but if it's scaled between 0 and 1, a value of 0.3257 seems very low. Wouldn't that very low modeling quality influence the interpretation of the results?

Line 216: Soft-DTW: is this a pre-processing step of the data, or is this already one of the clustering steps of the CSP? Unclear

Line 243-245: as we established above, 1 to 4 pixels basically cover the entirety of the Setinel-2 chip, so the lack of variability stems from the resampling of the dataset, not from the variables themselves.

Line 251: again wrong units? (km2?)

Line 251: given the spatial resolution of the meteorological dataset, what is the goal of first upscaling, then downscaling? Is the downscaled data used somewhere else?
Otherwise this step seems overkill to me, just take the average over the sentinel-2 extent.

Line 256-307: rephrase this section, the definitions are all over the place.

Figure 1: I'm not sure I understand the axis, the y-axis is all NDVI?

Line 326-334: introduce Si after equation 9, this is confusing.

Line 412: "irrespective of the weather segment the sensitivity of the variables remained almost the same". Does that mean that the clustering wasn't needed?

Line 414: "weighted by the cardinality of the sample sets". Not sure what that means.

# Minor comments

Figure 1: For each line, use same limits for y
Figure 2: use same limits for NDVI

# Grammar comments

Citations miss-match between automatic citations and manual:
117
131/132
145
... and many more

Author name twice:
107/108
113/114
119
128
134
... and many more

Line 24: makes it sound like EarthNet2021 is your previous study

Line 100: missing point between citation and reference

Line 101: "insights. In..." new sentence here
line 102: remove authors, "propose"

Line 128: replace "concerning" with "on"

Line 143: "during dry years, like 1989, following another dry year." unclear what that means

**Questions:**

c.f. weaknesses above.

---

### Official Review · Reviewer_8x9V · 2024-10-28

**Soundness:** 3
**Presentation:** 2
**Contribution:** 2
**Rating:** 5
**Confidence:** 4

**Summary:**

This paper discusses how to develop an interpretability pipeline for spatiotemporal prediction problems in earth science. The authors point out that existing interpretability methods, such as PDP or LIME, may have two limitations when dealing with spatiotemporal prediction problems:

1. These methods lack consideration of spatiotemporal variations in independent variables, and their randomly generated samples for explanatory analysis may lack temporal continuity.

2. These methods ignore that the relationships between independent and dependent variables may vary with seasons, lacking the ability to explain temporal dynamic mappings.

To address these issues, the authors propose an interpretability pipeline called CLUSTER-SEGREGATE-PERTURB (CSP). This method first uses Soft DWT+Kmeans to cluster the temporal variation curves of meteorological data into several patterns. Based on the patterns of meteorological attributes obtained from clustering, the authors divide the dataset into several subsets according to meteorological pattern similarity. Finally, the authors create perturbation sets for various variables based on climate patterns and data averages for subsequent interpretability analysis.

In the analysis phase, the authors selected three influence variables: temperature (min, max, avg), pressure, and precipitation. They trained a ConvLSTM model on the EarthNet21 dataset and used marginal sensitivity and marginal correlation analysis to interpret ConvLSTM on the segregation-perturbation sets generated by CSP. The authors analyzed the results of marginal sensitivity and marginal correlation. The results show that NDVI is most sensitive to precipitation, followed by temperature and pressure.

**Strengths:**

This paper demonstrates good originality. The authors' analysis of existing PDP and LIME methods is reasonable, and they accurately identify the scientific challenges in these two approaches. To address these issues, the authors propose a sound scientific hypothesis: "explaining grouped spatiotemporally homogeneous data." Based on this hypothesis, they construct a three-step pipeline using existing methods to aggregate spatiotemporally homogeneous data and add perturbations that preserve temporal continuity. Using this approach, the authors discuss the influence of three main meteorological factors on NDVI and draw some interesting conclusions.

**Weaknesses:**

However, the paper has significant issues in its research foundation and experimental validation. Additionally, there are shortcomings in the literature review and paper organization.

Research Foundation:
The paper primarily builds upon improving PDP and LIME methods. While these methods indeed show limitations in explaining time series predictions, recent studies have already extended these interpretability methods to time series prediction:

[1] Shi, H., Yang, N., Yang, X., & Tang, H. (2023). Clarifying relationship between pm2. 5 concentrations and spatiotemporal predictors using multi-way partial dependence plots. Remote Sensing, 15(2), 358.

[2] Liu, J., & Zhang, X. (2022). ReX: A Framework for Incorporating Temporal Information in Model-Agnostic Local Explanation Techniques. arXiv preprint arXiv:2209.03798.

The authors' failure to consider and analyze these methods undermines the reliability of their research foundation.

Method Validation:
Given that this is not the only post-hoc interpretation method for time series data, it is essential to validate whether this method is more suitable for surface process spatiotemporal prediction than existing methods. Specifically, the paper lacks discussion of:

1. Quantitative evaluation of interpretability method accuracy (typically achieved using surface process simulation synthetic datasets)
2. Comparative analysis with other time series interpretation methods
3. Response to research hypotheses regarding existing methods' reliability with temporal and high-dimensional data

Additional Issues:
1. The related work section merely lists existing methods without analyzing their limitations or providing technical motivation

2. The paper's structure is overly segmented, mixing original methodology (5.2), previous research methods (5.1), and experimental results (5.2)

3. While claiming existing methods struggle with high-dimensional spatiotemporal data (lines 21 and 52), the experiments only use data with 3 attributes and 5 dimensions

4. There's an inconsistency between the criticism of PDP's inability to isolate variable effects (line 53) and the paper's assumption of variable independence in Section 4

**Questions:**

1. What is the current state of spatiotemporal interpretability methods, and what challenges do they face?

2. What is the motivation behind the proposed CSP technique, and what implications does it have for future research?

3. What are the technical advantages of the proposed method compared to existing spatiotemporal interpretability methods, and is it more suitable for surface process spatiotemporal prediction problems?

4. How reasonable is the assumption of independent distribution among meteorological variables? In reality, meteorological variables typically exhibit significant correlations.

5. Can the authors provide validation results on synthetic datasets and quantitatively compare the accuracy of their method with other time series interpretability methods?

6. NDVI may experience saturation effects under strong solar radiation. Does the interpretability model account for the instability in independent-dependent variable relationships caused by this saturation phenomenon?

7. Can the authors demonstrate their method's performance on truly high-dimensional data to validate its advantages over existing methods in handling high-dimensional data?

8. Is there an issue with the y-axis labeling in Figure 1?

---

### Official Review · Reviewer_hCsD · 2024-11-03

**Soundness:** 2
**Presentation:** 2
**Contribution:** 1
**Rating:** 3
**Confidence:** 4

**Summary:**

This work tries to uncover the relationship of meteorological drivers to satellite-derived vegetation greenness through studying the partial sensitivities of a deep neural network to input perturbations. More specifically, they work on the EarthNet2021 dataset, which contains samples ("minicubes") of high resolution NDVI at many places in Europe co-located with meteorological variables. First, they cluster each individual meteorological variable. Then they identify similar minicubes, by grouping according to the cluster ids of five individual meteorological variables, resulting in ~800 meteorologically similar minicubes. For each group, they compute partial sensitivities of the ConvLSTM model to additive perturbations of the input meteorology, obtaining marginal response curves of NDVI to weather.

**Strengths:**

1) The work is original in the sense that it tried working around the challenges the particular dataset (being high-dimensional and not global, and samples auto-correlated) brings, leading a highly tailored analysis for the EarthNet2021 dataset.
2) Reproducibility is given, as the work is based on open data and open source code of a model.
3) The paper highlights the challenging nature of the relationship between NDVI and weather, often being non-linear, and highly variable in space.

**Weaknesses:**

1) My major concern is that the presented analysis does not achieve its goal, that is explaining the complex relationship between NDVI and weather at fine resolution. Here, i think a more careful presentation and interpretation of the results needs to be done, after reading this paper, the reader should have an increased understanding of the drivers of NDVI dynamics.
2) The used dataset EarthNet2021 and the Diaconu ConvLSTM models both suffer from a few drawbacks. It would have been much better to use the more recent GreenEarthNet dataset, which is an improved version of EarthNet2021 and comes with many improved baseline models. https://openaccess.thecvf.com/content/CVPR2024/html/Benson_Multi-modal_Learning_for_Geospatial_Vegetation_Forecasting_CVPR_2024_paper.html
3) The writing lacks clarity, I had to read the paper multiple times to understand the approach. Perhaps a figure that visually explains the approach could help here, but more importantly, many parts of the paper need to be rewritten to follow a clearer structure.
4) I am unsure if the clustering approach is meaningful for Precipitation data, which is generally exponentially distributed and has few spikes and else zeros.
5) Introducing around ~800 groups of minicubes with similar weather does not necessarily increase interpretability, as one would still need to look at 800 different plots with 5 panels each (Fig. 2...) to fully understand the relationship between NDVI and weather.
6) The perturbations are done univariate, however the considered weather variables are not independent of each other.
7) The perturbations for precipitation seem physically not meaningful.
8) The related works section is poorly written. It should reflect what has been done and how that related to the presented study. Also, it lacks many related works. You could start by checking the references of the EarthNet2021 paper, which already contains many more especially regarding NDVI.
9) In Fig. 1, I am not sure the absolute x-coordinate makes so much sense. The EarthNet2021 minicubes start at random start dates, i.e. they are during different seasons, so patterns in meteorology should rather be translation-equivariant, i.e. considering relative coordinates, I am thinking in the direction of wavelets and other filters here.

**Questions:**

1) I wonder if you could use climate analogues instead of the Perturbations? --> What if you switch the weather of one group of minicubes with that of another to compute the sensitivities? In that way you'd be more sure it is actually "physical" climate you are considering.
2) How do you match the resolution of the meteorological variables to Sentinel 2? Do you take a central cut-out that actually reflects the Sentinel 2 scale? Or do you interpolate, as in the spatial references do not match anymore?
3) Related to this, for the spatial averaging to do clustering of the meterological variables, are those time series representing 2.5km^2 or 100km^2?
4) Could you actually analyse if the relationships are different for different land cover types? E.g. Grasses and Trees should have very different relationships to weather, for instance under drought grasses can be brown within a few days, but evergreen trees take months before you visibly see changes in the NDVI.
5) The ConvLSTM uses memory, which may reflect some of the relationship between temperature and NDVI on a monthly time scale, in other words, the model might incorporate a trend for its meteorological variables or the response thereof in its memory - could that not potentially distort your analysis?
6) How do you account for lagged effects? The land surface can be much slower than the atmosphere, responding on very different time scales (slowly changing...)
7) Have you seen this recent pre-print? https://arxiv.org/abs/2410.01770
8) More of a suggestion, I personally feel like the contribution in this paper may be much more on the "results" side of things, and less methodological. Hence, it might be much more suitable for a journal and not for ICLR.

---

### Official Review · Reviewer_8GJb · 2024-11-04

**Soundness:** 3
**Presentation:** 2
**Contribution:** 1
**Rating:** 3
**Confidence:** 4

**Summary:**

This paper introduces the Cluster-Segregate-Perturb (CSP) pipeline as an approach for achieving explainability in land surface forecasting. The CSP pipeline includes clustering, segregation, and perturbation. This pipeline facilitates analyses such as marginal sensitivity, correlation, and lag analysis, etc for complex land forecasting models. CSP has been tested on EarthNet2021.

**Strengths:**

1. The idea of designing a pipeline for the complex spatiotemporal data and high-dimensional feature spaces is intriguing.
2. The description of the challenges in explainability of spatiotemporal land surface forecasting is technically sound.

**Weaknesses:**

1. The paper lacks novelty in its method design. The author may overstate the ability of CSP, for example, downsampling the images by averaging down to a single value is kind of rough. While it greatly reduces the complexity, it also loses the spatial information.
2. The paper’s presentation needs to be improved; the structure lacks coherence and unclear.
3.EARTHNET2021 is the only dataset being used. It is necessary to evaluate the method across various datasets in land surface forecasting fields. Also, consider using synthetic data to prove the functionality of CSP.
4. The experiments section mainly focuses on sensitivity and correlation analysis, lacking deeper and more detailed analysis. Additionally, there are no baselines for comparison.
5. The assumption of this method is not universally valid. Ignoring the relationships between input features may oversimplify the analysis since weather features are often correlated.

**Questions:**

1.	Could you elaborate on your method’s contribution and explain how it outperforms existing methods? Have you compared CSP to other approaches?
2.	Though you claim CSP can be integrated with any other model architecture in the paper, have you tried any other models except for ConvLSTM? If so, do you have the corresponding results?
3.	Have you evaluated CSP in scenarios where input features are correlated and do not satisfy the assumption?

---

### Note · Authors · 2024-11-13

I have read and agree with the venue's withdrawal policy on behalf of myself and my co-authors.